### elife.elifesciences.org

# Fluorescent sensors for activity and regulation of the nitrate transceptor CHL1/NRT1.1 and oligopeptide transporters

**Cheng-Hsun Ho, Wolf B Frommer***

Department of Plant Biology, Carnegie Institution for Science, Stanford, United States

**Abstract** To monitor nitrate and peptide transport activity in vivo, we converted the dual-affinity nitrate transceptor CHL1/NRT1.1/NPF6.3 and four related oligopeptide transporters PTR1, 2, 4, and 5 into fluorescence activity sensors (NiTrac1, PepTrac). Substrate addition to yeast expressing transporter fusions with yellow fluorescent protein and mCerulean triggered substrate-dependent donor quenching or resonance energy transfer. Fluorescence changes were nitrate/peptide-specific, respectively. Like CHL1, NiTrac1 had biphasic kinetics. Mutation of T101A eliminated high-affinity transport and blocked the fluorescence response to low nitrate. NiTrac was used for characterizing side chains considered important for substrate interaction, proton coupling, and regulation. We observed a striking correlation between transport activity and sensor output. Coexpression of NiTrac with known calcineurin-like proteins (CBL1, 9; CIPK23) and candidates identified in an interactome screen (CBL1, KT2, WNKinase 8) blocked NiTrac1 responses, demonstrating the suitability for in vivo analysis of activity and regulation. The new technology is applicable in plant and medical research.

**\*For correspondence:**
wfrommer@carnegiescience.edu

**Competing interests:** The authors declare that no competing interests exist.

**Reviewing editor**: Richard Aldrich, The University of Texas at Austin, United States

## Introduction

Quantitatively, nitrogen is the single most limiting nutrient for plants. Thus, not surprisingly, maximal crop yield depends critically on nitrogen fertilizer inputs. Current practices require production of ~$1.5 \times 10^7$ tons of N-fertilizer per annum, consuming ~1% of the world's annual energy production. Plants absorb only a fraction of the fertilizer applied to the field, leading to leaching into groundwater, polluting the environment, and damaging human health. Improvements in nitrogen use efficiency of crops are urgently required; although potential targets including uptake transporters and metabolic enzymes have been identified, successful improvements in N-efficiency are rare (*McAllister et al., 2012*; *Jiang, 2012b*; *Xu et al., 2012*; *Schroeder et al., 2013*). Overexpression of an alanine amino transferase or the transporter *OsPTR9* are two of the few examples of improved nitrogen use efficiency (*Shrawat et al., 2008*; *Fang et al., 2013*). Ammonium, nitrate, amino acids, and di- and tripeptides serve as the major forms of inorganic and organic nitrogen for plants. Uptake occurs predominantly from the soil/rhizosphere into roots, although aerial parts of the plant are also capable of absorbing nitrogen (*McAllister et al., 2012*). Nitrogen availability and distribution in soil vary both spatially and temporally. Inorganic nitrogen uptake is complex and involves multiple ammonium and nitrate uptake systems, typically grouped into low-affinity/high-capacity and high-affinity/low-capacity systems (*Siddiqi et al., 1990*; *Wang et al., 1994*; *von Wirén et al., 1997*). Their relative activity is influenced by both exogenous and endogenous factors. The exact sites of uptake of the various forms of nitrogen along the length of the root, the cells that are directly involved, and in vivo regulation are not well understood. Also, the exact intercellular path towards the stele is not experimentally proven. The reasons for this lack of knowledge lie in the

**eLife digest** About 1% of global energy output is used to produce nitrogen-enriched fertiliser to improve crop yields, but much of this energy is wasted because plants absorb only a fraction of the nitrogen that is applied as fertiliser. Even worse, the excess nitrogen leaches into water sources, poisoning the environment and causing health problems. However, to date, most efforts to increase the efficiency of nitrogen uptake in plants have been unsuccessful.

The key to improving the uptake efficiency of a nutrient is to identify obstacles in its journey from the soil to cells inside the plant. The first obstacle that nitrate ions encounter is the membrane of the cells on the surface of the roots of the plant. Many researchers believe that it would be possible to increase the amount of nitrogen absorbed by the plant if more was known about the ways that plants control how nitrate ions and other chemicals enter cells.

The cell membrane contains gated pores called transporters that allow particular molecules to pass through it. Although the transporters responsible for the uptake of nitrate ions, peptides, and ammonium ions (the main nitrogen compounds that plants acquire) have been identified, current experimental techniques cannot determine when and where a specific transporter is active within a living plant. This makes it difficult to know where to target modifications and to determine how effective they have been at each step. The nitrate transporter also acts as an antenna that measures nitrate concentration to ensure it is used optimally in the plant, but current techniques cannot show how this actually works.

Now, Ho and Frommer have exploited the fact that a transporter changes shape as it does its job to create sensors that can track the movement of nitrate and peptides through the cell membrane. By using fluorescent proteins to monitor how the shape of the transporter changes, Ho and Frommer were able to measure how structural mutations and regulatory proteins influenced the movement of nitrate and peptides through the membrane.

For efficiency, all of this work was performed in yeast cells. The next goal is to use the technique in plants to uncover how they adjust to changes in nutrient levels in the soil.

fact that nitrogen transport is difficult to measure. Some studies rely on the analysis of the depletion of the medium, others use stable isotopes, or the $^{13}$N-isotope, which has a short half-life time of ~10 min and requires access to a suitable supply source (*Wang et al., 1993*; *Clarkson et al., 1996*). Most of these techniques lack spatial resolution, that is information on which cell layers and which root zones absorb the nutrient. Electrophysiological assays can provide spatial information; however, they are mostly used at accessible surfaces. Spatial information has been provided in a few studies by methods such as vibrating electrodes (*Henriksen et al., 1990*, *1992*), positron-emitting tracer imaging systems (*Matsunami et al., 1999*; *Kiyomiya et al., 2001*), or secondary ion mass spectrometers (*Clode et al., 2009*). We also know little about differences in the distribution of the nitrogen forms in different root cell types or zones and with respect to cellular compartmentation. Classical approaches average total ion/metabolite levels over all cells in the sample, for example, in whole roots. Nitrate levels differ dramatically between root cell types (*Zhen et al., 1991*; *Karley et al., 2000*). Recently, a GFP-labeled protoplast-sorting platform was used to compare metabolomes of individual cell types in roots (*Moussaieff et al., 2013*). This study found that the levels of small oligopeptides were comparatively higher in the epidermis and endodermis compared to other root cell types. Compartmental analyses indicated that the nitrate concentration of root vacuoles is ~10-fold higher compared to the cytosol (*Zhen et al., 1991*).

Transporters are placed in strategic positions to control which and how much of a specific nitrogen form can enter a given cell at a given point of time. The progress in identifying transporter genes provided a new handle for addressing the mechanisms and the spatial and temporal regulation of nitrogen acquisition from a new level of detail. Three major families of transporters for inorganic nitrogen uptake (and distribution) have been identified: the NPF/POT nitrate transporter family (*Leran et al., 2013*), NRT2 nitrate transporters (*Kotur et al., 2012*), and the ammonium transporters of the AMT/MEP/Rh family (*von Wirén et al., 2000*; *Andrade and Einsle, 2007*). In addition to their role in nitrate uptake, members of the NPF/POT (*Leran et al., 2013*) family play important roles also in the transport of histidine, dicarboxylates, oligopeptides, and glucosinolates, and surprisingly at least three major plant

hormones: auxin, ABA, and gibberellin (*Krouk et al., 2010*; *Kanno et al., 2012*; *Boursiac et al., 2013*). Genes are valuable tools for exploring physiological functions. Analysis of RNA levels allows us to study gene regulation (*Gazzarrini et al., 1999*), for example, transcriptional GUS-fusions for determining organ and cell type specific expression and translational GFP-fusions for subcellular localization. Both classical and novel methods, including cell-specific transcriptional profiles and 'translatomes' provide us with new insights into differences in the expression of transporters in roots (*Brady et al., 2007*; *Mustroph et al., 2009*). Analysis of cell type-specific expression profiles showed that the majority of changes in nitrate-induced gene expression are cell-specific (*Gifford et al., 2008*). Expression and purification of the proteins followed by reconstitution in vesicles or expression in heterologous systems to interrogate biochemical properties include $K_m$ and transport mechanisms. The genes can be used as a basis for structure function studies (*Loqué et al., 2007*) and to obtain crystal structures (*Andrade et al., 2005*; *Doki et al., 2013*). We can use the genes to identify interacting proteins (*Lalonde et al., 2010*). Importantly, the availability of genes enables us to generate specific mutants (*Yuan et al., 2007*; *Wang et al., 2009*) that provide insights into their physiological roles. However, even with this massive amount of detailed data, the key information is missing, namely the information on the activity state of a given protein in vivo. In vivo activity depends mainly on two additional parameters beyond protein abundance at a given membrane: the local concentration of the substrate/s, the status of the cell (e.g., the membrane potential and local pH as key determinants for ion transporter activity), and the status of cellular regulatory networks required for the activity of the protein in question. Again, genes can help us to find regulators and study the effect of mutations on nitrogen acquisition, but ultimately we need to be able to quantify the activity of the transporters in individual cells in vivo.

Nitrogen uptake is controlled by many factors, such as nitrogen level, energy status of the plant, assimilation status of imported nitrogen, N-demand, and involved mobile signals between shoots and roots as well as between different parts of the root system (*von Wirén et al., 1997*). Nitrate transporters are regulated through phosphorylation, mediated by calcium-dependent calcineurin-like kinases (Calcineurin B-like, CBL and CBL-interacting protein kinase, CIPK) (*Ho et al., 2009*; *Hu et al., 2009*; *Wang et al., 2009*). Major breakthroughs were findings that indicate both the members of the AMT and NPF/POT family function as transporters and receptors (transceptors) (*Ho et al., 2009*; *Lima et al., 2010*; *Rubio-Texeira et al., 2010*). However, despite broad progress, at present, we have only a limited understanding of signaling pathways that control nitrogen acquisition.

It is important to develop tools for monitoring the activity of individual transporters in specific locations in individual cells of plant roots in a minimally invasive manner. A minimally invasive tool that has proven valuable for monitoring ions and metabolite levels with high spatial and temporal resolution is genetically encoded fluorescent nanosensors (*Okumoto, 2012*). These sensors rely on substrate binding–dependent conformational rearrangements in a sensory domain. The rearrangements are reported by changes in Förster Resonance Energy Transfer (FRET) efficiency between two fluorescent proteins, which act as FRET donor and acceptor due to spectral overlap. Sensors for glucose, sucrose, and zinc have successfully been used in Arabidopsis to monitor steady state levels as well as accumulation and elimination under both static and dynamic conditions where roots were exposed to pulses of the respective analytes (*Deuschle et al., 2006*; *Chaudhuri et al., 2008*; *Okumoto et al., 2008*; *Chaudhuri et al., 2011*; *Lanquar et al., 2014*).

The recent progress in obtaining crystal structures for transporters, and more importantly the availability of transporter structures in multiple configurations, has provided insights into the conformational rearrangements occurring during the transport cycle (*Doki et al., 2013*; *Guettou et al., 2013*; *Henderson and Baldwin, 2013*; *Madej et al., 2013*). Biochemical and structural analyses have shown that many transporters undergo conformational changes during the transport cycle (*Shimamura et al., 2010*; *Jiang, 2012a*; *Krishnamurthy and Gouaux, 2012*). Important in this context is that such rearrangements have been observed for many members of the MFS superfamily, including members of the NPF/POT family (*Doki et al., 2013*). We therefore hypothesized that it should be possible to 'record' the conformational rearrangements that occur during the transport cycle in a similar manner as used for the engineering of the FRET sensors. The first prototype for transport activity sensors, named AmTrac, uses ammonium transporters as sensory domains for engineering transport activity sensors by inserting a circularly permuted EGFP (cpEGFP) into a conformation-sensitive position of an ammonium transporter (*De Michele et al., 2013*). Addition of ammonium to yeast cells expressing the AmTrac sensor triggers concentration-dependent and reversible changes in fluorescence intensity (*De Michele et al., 2013*). Whether this approach is transferable to other family proteins in different species remained to

be shown. To create nitrate and peptide transport activity sensors, we fused CHL1 and four PTRs to fluorescent protein pairs, expressed the fusions in yeast, and tested their response to substrate addition (named NiTrac for nitrate transport activity and PepTrac for peptide transport activity). The five sensors responded to the addition of nitrate or peptides, respectively. The kinetics of the NiTrac1 sensor response was strikingly similar to the transport kinetics of the native CHL1; the response was specific and reversible. The new sensors were used to study structure/function relationships, to correlate effects of mutations in CHL1 and NiTrac1 on activity and sensor responses, and to observe the effect of potential regulators on the conformation of the transporter. The successful use of the sensors in yeast indicates that these new tools can be used for *in planta* analyses.

## Results

### Engineering of a nitrate transport activity sensor

It is likely that the nitrate transceptor CHL1 undergoes conformational rearrangements during its transport cycle. To measure substrate-dependent conformational rearrangements, CHL1 was sandwiched between a yellow acceptor (Aphrodite) and cyan donor fluorophore (mCerulean) (*Rizzo et al., 2006*; *Figure 1A*). This chimera, named NiTrac1, was expressed in yeast, followed by spectral analysis of yeast cultures in a spectrofluorimeter (*Figure 1B*). The fluorophores were in Förster distance, as evidenced by significant resonance energy transfer. If conformational rearrangements were induced by substrate addition, one might expect a change in the energy transfer rate. To our surprise, and in contrast to typical FRET sensors (e.g., glucose or glutamate [*Fehr et al., 2003*; *Okumoto et al., 2005*]), we observed an overall reduction in the emission intensities of both donor and acceptor, but no obvious change in FRET efficiency. Cyan FPs are typically robust compare to the yellow variants; specifically, they are less sensitive to pH changes or other ions compared to yellow variants (here Venus encoded by codon-modified Aphrodite gene sequence [*Deuschle et al., 2006*]). However, Aphrodite emission was unaffected by nitrate when excited directly (*Figure 1B*, inset), indicating that the external nitrate triggers donor quenching in the cytosol. The sensor response can be expressed as a ratio change between the emission intensity of the sensor at CFP excitation relative to YFP emission obtained from acceptor excitation. As one may have expected, the nitrate analog chlorate that lead to the naming of CHL1 (chlorate resistance of the *chl1* mutant) (*Tsay, 1993*) also triggered NiTrac quenching (*Figure 1C*). The response of NiTrac1 is nitrate- and chlorate-specific; other compounds such as chloride, ammonium, divalent cations, and dipeptide had no significant effect (*Figure 1D*). When mCerulean was replaced by the corral-derived cyan fluorescent protein mTFP (*Ai et al., 2008*), we observed FRET, but nitrate addition had no effect on the emission of this variant (*Figure 1E*). The mTFP variant, named NiTrac1c (control), therefore can serve as a control sensor for in vivo measurements. Replacement of mCerulean with eCFP, another jellyfish variant, retained the donor-quenching response to nitrate (*Figure 1F*). Although we do not understand the mechanism by which nitrate triggers donor quenching, the effect is likely related to a specific property common to mCerulean and eCFP and lacking in mTFP.

### Engineering of four peptide transport activity sensors

It is conceivable that nitrate is taken up by CHL1 into the cytosol where it binds to mCerulean or eCFP, leading to quenching. However, addition of nitrate to yeast cells expressing CHL1 alone had no effect on the fluorescence of a cytosolically expressed mCerulean (*Figure 1G*). One could argue that quenching occurs locally at the exit pore of the transporter directly at the plasma membrane and thus requires tethering of mCerulean to the transporter. To test whether quenching is specifically caused by nitrate, we created similar constructs for the oligopeptide transporters PTR1, 2, 4, and 5 from Arabidopsis (*Komarova et al., 2012*; *Tsay et al., 2007*; *Leran et al., 2013*; *Figure 2A*). These proteins share between 39% and 74% homology with CHL1. PepTrac1, PepTrac2, and PepTrac5 sensors all responded with donor quenching to the addition of 0.5 mM diglycine (*Figure 2B–D*). Interestingly, PepTrac4 responded to substrate addition with a ratio change that is consistent with a change in the energy transfer rate rather than donor quenching (*Figure 2E*). Further characterization will be necessary to explore the molecular basis of donor quenching and how conformational rearrangements cause donor quenching in NiTrac1 by nitrate or PepTrac1 by peptides and how they induce resonance energy transfer in PepTrac4.

### Biphasic kinetics of the NiTrac1 response

The conformational rearrangements in the sensors could be induced by substrate binding or reflect rearrangements that occur during the transport cycle. Because binding and transport typically have

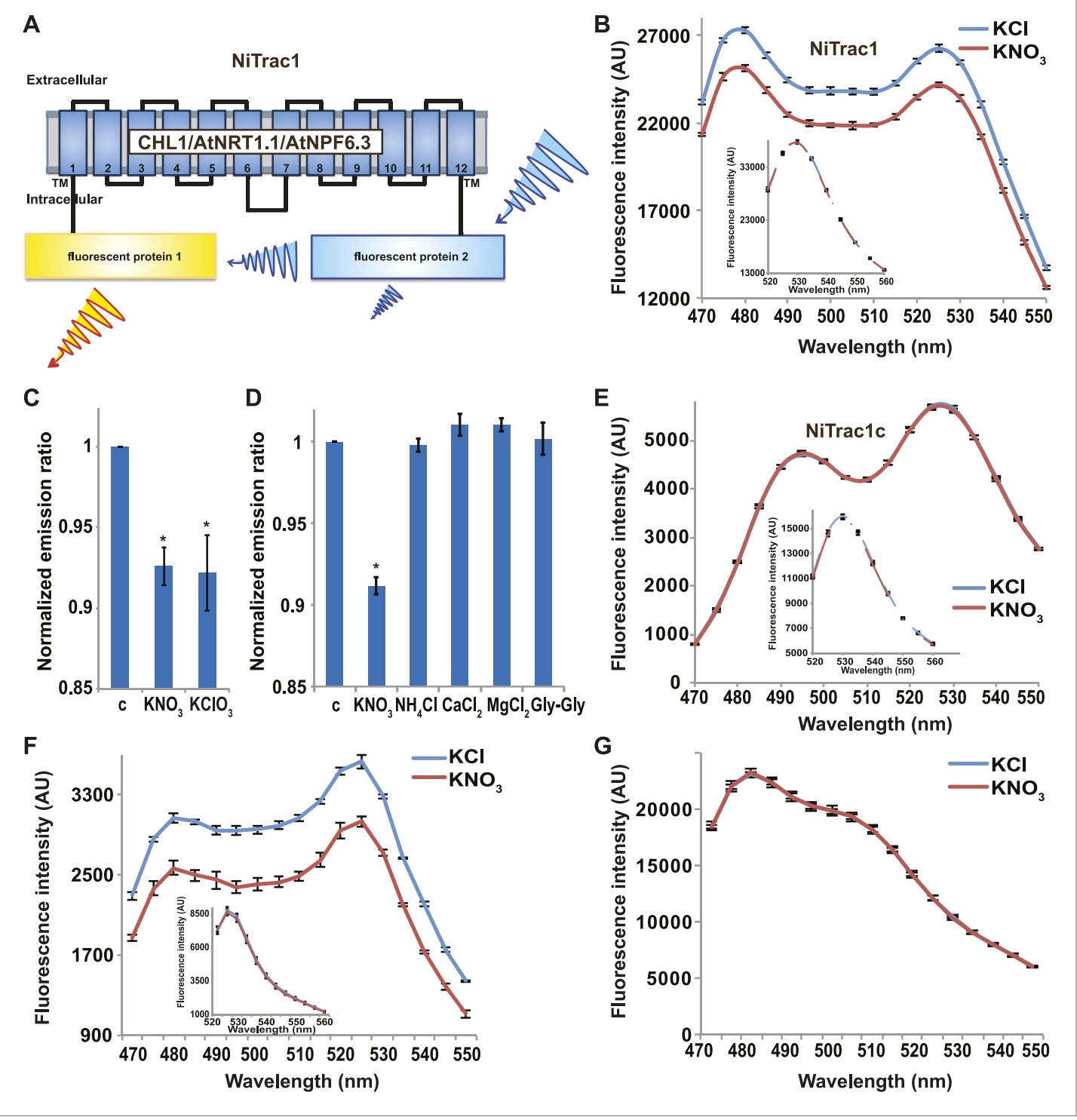

**Figure 1.** Design and development of NiTrac sensors. (**A**) Schematic representation of the NiTrac1 sensor construct. Aphrodite, yellow; mCerulean, light blue; CHL1/NRT1.1/NPF6.3 dark blue; TM, transmembrane domain. (**B**) Emission spectra for NiTrac1 expressed in yeast cells; excitation at 428 nm: addition of 5 mM potassium nitrate (red; control 5 mM KCl, blue) lead to a reduction in fluorescence intensity of donor and acceptor emission, caused by donor quenching. Inset: emission of Aphrodite in NiTrac1 when excited at 505 nm. Aphrodite emission was unaffected. (**C**) Nitrate and its analog chlorate both trigger quenching at 5 mM concentrations. Nitrate-induced ratio change (peak fluorescence intensity of Aphrodite excited at 505 nm over emission spectrum at 485 nm obtained with excitation at 428 nm). Data are normalized to buffer-treated control c. (**D**) Substrate specificity: yeast cells expressing NiTrac1 were treated with the indicated compounds at 5 mM concentrations. Only nitrate and chlorate triggered responses that were

*Figure 1. Continued on next page*

*Figure 1. Continued*

significantly different from control c (*p<0.05, *t* test). Experiment performed as in *Figure 1C*. (**E**) Absence of quenching of NiTrac1 when mCerulean was exchanged for mTFP (excitation at 440 nm). Inset: emission of Aphrodite in NiTrac1 when excited at 505 nm. (**F**) Donor quenching is retained when mCerulean is exchanged for eCFP in NiTrac1 in response to addition of 5 mM potassium nitrate (red; control 5 mM KCl, blue; excitation at 428 nm). Inset: emission of Aphrodite in NiTrac1 when excited at 505 nm. (**G**) No detectable effect on the fluorescence properties of nitrate addition to yeast cells coexpressing a cytosolically localized free mCerulean and the CHL1 transceptor. Mean ± SD; n = 3.

different kinetic constants, we analyzed the response kinetics of NiTrac1. CHL1 is unusual in that it shows biphasic nitrate uptake kinetics (*Figure 3A*; *Liu and Tsay, 2003*). The observed dual-affinity in oocytes had been attributed to phosphorylation of T101 by endogenous kinase (*Liu and Tsay, 2003*). The phosphorylation hypothesis would suggest that about half of the transporter molecules are phosphorylated. Interestingly, we observed that the kinetics of the fluorescence response of NiTrac1 in yeast were also biphasic (*Figure 3A*). Since it is unlikely that yeast also partially phosphorylates the transporter, the biphasic kinetics are more likely an intrinsic property of the protein. Mutation of T101 to alanine had been shown to eliminate the high-affinity component (*Figure 3B*; *Liu and Tsay, 2003*). Introduction of T101A into NiTrac1 also eliminated the high-affinity component, intimating that NiTrac1 is a transport activity sensor, and that conformational rearrangements during the transport cycle affect mCerulean emission (*Figure 3B*). Interestingly, the transport $K_m$s of both high- and low-affinity phases matched the values obtained for the fluorescence response, supporting the hypothesis that NiTrac measures the transport activity. Measurement of the sensor response in individual yeast cells demonstrated rapid nitrate-induced quenching and reversibility of the fluorescence intensity after removal of nitrate (*Figure 3C*), indicating that the sensor can be used effectively for *in planta* analyses.

## Effect of mutations on the NiTrac1 response and NRT1.1 activity

To study the NiTrac mechanism in more detail and to identify residues important for the transport function of the transporter and sensor, we generated a homology model for CHL1 on the basis of crystal structures of bacterial proton-dependent oligopeptide transporter homologs ('Materials and methods') and predicted potentially functionally important residues structurally close to the substrate binding pocket from the predicted structure and from sequence alignments. We specifically targeted residues that might be important for substrate specificity, residues involved in proton cotransport, and salt bridges possibly involved in dynamic movements during the transport cycle (*Figure 4A*). As one may have expected, different mutants showed different energy transfer ratios, consistent with conformational differences (altered distance and/or orientation of the fluorophores in the absence of substrate; *Figure 4B*). Interestingly, we not only observed cases in which donor quenching was lost, but also changes that are consistent with changes in FRET efficiency in response to ligand addition, as well as mixtures of donor quenching and change in the FRET efficiency (*Figure 4B*). However, without knowledge of the effect of the mutations on transport activity, the data are difficult to interpret. Therefore, we introduced the corresponding mutations into CHL1, expressed the mutants in *Xenopus* oocytes and used two-electrode voltage clamp (TEVC, *Figure 5*), and $^{15}$N-uptake (*Figure 6*) to measure the transport activity. In response to nitrate addition, CHL1 expressing oocytes showed an inward current, consistent with the proposed $2H^+/NO_3^-$ cotransport mechanism. CHL1 contains a highly conserved motif E41-E44-R45 in TM1 predicted to play a role in proton coupling (*Newstead, 2011*; *Newstead et al., 2011*; *Solcan et al., 2012*; *Doki et al., 2013*). Mutations in this motif in the oligopeptide transporters PepT$_{St}$ (from *Streptococcus thermophiles* [*Solcan et al., 2012*]), PepT$_{So}$, PepT$_{So2}$ (both from *Shewanella oneidensis*) (*Newstead, 2011*), and GkPOT (from *Geobacillus kaustophilus*) (*Doki et al., 2013*) typically lost proton-driven transport activity. We therefore tested the role of residues in this motif using NiTrac expressed in yeast and CHL1 expressed on oocytes. Mutation in any of the three residues (E41A, E44A, and R45A, TM1) led to a loss of nitrate-induced currents and $^{15}$N-uptake in both the high- and low-affinity range (0.5/0.25 and 10 mM) (*Figures 5, 6*). The corresponding mutant of NiTrac1 also lost the sensor response to nitrate addition (*Figures 4B, 7*), indicating that the conserved motif is also used for proton cotransport of nitrate. Interestingly, the mutant was characterized by higher FRET compared to wild-type CHL1, indicating that the mutation leads to a conformational change in the protein (*Figure 4B*). Structural and functional analyses of the bacterial peptide transporter PepT$_{St}$ had implicated a salt bridge between a conserved K126 in TM4 and E400 in TM10 in peptide recognition and/or structural movements during the transport cycle (*Solcan et al., 2012*). This lysine is conserved

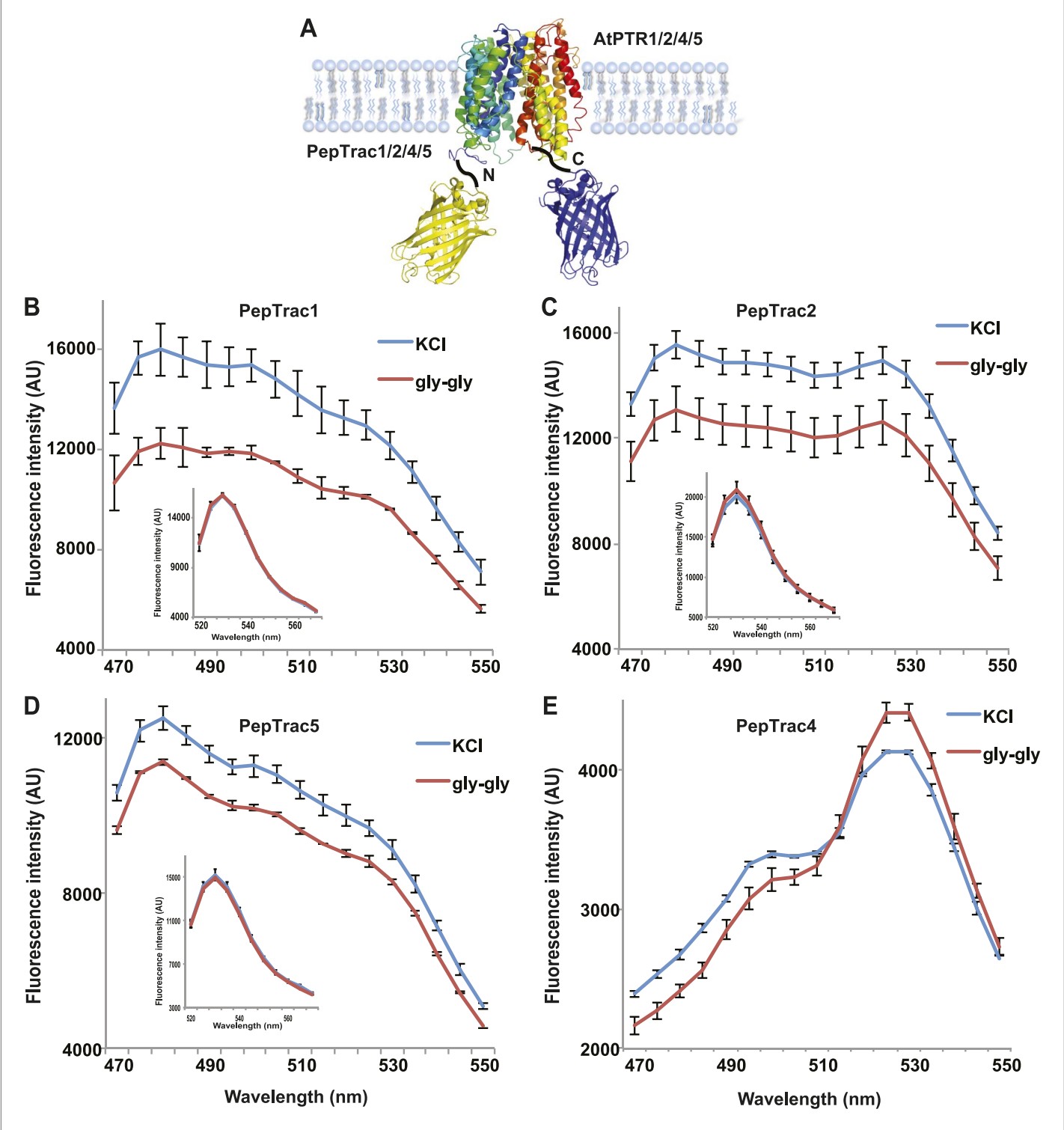

Figure 2. PepTrac sensors. (A) Schematic representation of the PepTrac sensor constructs. AtPTR1, 2, 4, and 5 were used for PepTrac sensor creation. Three-dimensional model of AFP-PTRs-mCerulean chimeric protein based on the crystal structure of bacteria peptide transporters ('Materials and methods'). PTR1 is shown in rainbow cartoon; AFP in yellow; mCerulean in blue. (B–D) Donor quenching of PepTrac1, 2, and 5 expressed in yeast in response to addition of 0.5 mM diglycine (red; control 5 mM KCl, blue; excitation at 428 nm). Inset: emission of Aphrodite in PepTrac1, 2, and 5 when excited at 505 nm. (E) FRET ratio change for PepTrac4 (red; control 5 mM KCl, blue; excitation at 428 nm). Mean ± SD; n = 3.

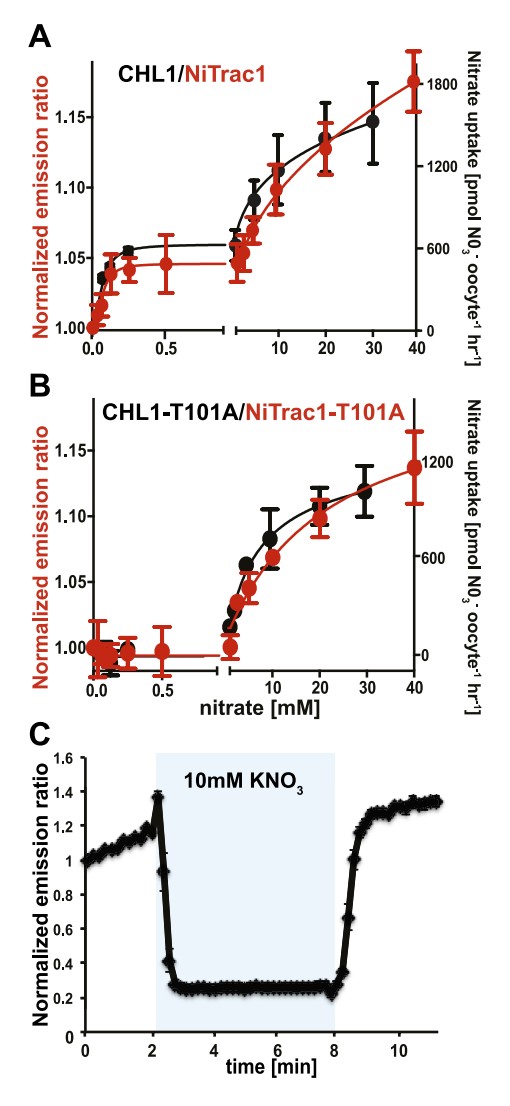

**Figure 3**. Biphasic kinetics of the NiTrac1 response. (**A**) Biphasic nitrate uptake kinetics of the fluorescence response of NiTrac1 (red) and biphasic nitrate uptake transport kinetics of CHL1/NRT1.1 (Black). (**B**) Monophasic nitrate uptake kinetics of the fluorescence response of NiTrac1-T101A (red) and monophasic low-affinity transport kinetics of CHL1/NRT1.1-T101A (Black, oocyte uptake data from (**Liu and Tsay, 2003**). The $K_m$s of NiTrac1 for nitrate are ~75.1 ± 21 μM and 3.8 ± 2.6 mM; for NiTrac1-T101A is 3.5 ± 3.7 mM. Excitation and emission as **Figure 1C**. The amount of decreased fluorescence intensity by addition of indicated nitrate concentration in **Figure 3A,B** were normalized to water-treated control (0) (mean ± SD; n = 3). (**C**) Analysis of the NiTrac1 response in individual yeast cells trapped in a Cellasic microfluidic plate. Cells were initially perfused with 50 mM MES buffer pH 5.5, followed by a square pulse of 10 mM $KNO_3$ in MES buffer for 6 min (blue frame). Data were normalized to the initial value (mean ± SD; n = 3).

throughout the POT family (K164 of CHL1, TM4) (**Newstead, 2011**; **Newstead et al., 2011**; **Solcan et al., 2012**; **Doki et al., 2013**). Consistent with results from the bacterial PepT$_{St}$ and GkPOT, mutation of K164 to alanine or aspartate in CHL1 completely abolished nitrate uptake in both the high- and low-affinity range (0.25 and 10 mM) (**Figure 6**); however, the nitrate-dependent inward currents were retained (**Figure 5**). Mutation K136A in GkPOT and K126A in PepT$_{St}$ both abolished completely proton-driven uptake but still had counterflow activities (**Doki et al., 2013**; **Solcan et al., 2012**). Both CHL1-K164 mutants either function as nitrate-dependent proton channels or have lost selectivity, and consistent with the shift of the reversal potential to more negative values transport other anions such as chloride. NiTrac1-K164A surprisingly showed a different response mode, that is upon addition of nitrate the mutant not only showed donor quenching but apparently also a change in FRET efficiency, underlining the exceptional sensitivity of NiTrac1 to effects of mutations on conformation (**Figure 4B**). Mutation of the salt bridge acceptor E476A in TM10 of CHL1 led to loss of both the nitrate-induced inward current and $^{15}$N-uptake (**Figures 5, 6**) and NiTrac lost the sensor response to addition of nitrate (**Figure 4B**); by contrast, and as one might expect, the conservative mutation E476D had no significant effect on transport properties and sensor response (**Figures 4B, 5, and 6**). Alanine substitutions were introduced into corresponding sites predicted to be in the vicinity of the substrate-binding pocket (L49, Q358, and Y388 in TM1, TM7, and TM8, respectively). Consistent with the results obtained for the corresponding residue (N342 in TM8) in GKPOT (**Doki et al., 2013**), Y388A had no detectable effect on the nitrate-induced inward currents, $^{15}$N-uptake, and sensor response (**Figures 4B, 5, and 6**), indicating the residue Y388 may not be involved in nitrate binding or transport cycle of CHL1. Mutation of L49 in TM1 and Q358 in TM7 of CHL1 to alanine had no significant effect on nitrate-induced inward currents and $^{15}$N-uptake (**Figures 5, 6**), but NiTrac responses were characterized by a mixture of donor quenching and FRET change (**Figure 4B**). Based on the protein sequence alignments, CHL1 carries an extended cytoplasmic loop connecting the N- and C-terminal six helical bundles. To test the role of this loop, a triple mutant E264A-E266A-K267A was analyzed. The triple mutant lost specifically the low-affinity component nitrate-induced inward current and $^{15}$N uptake but retained the high-affinity component (**Figures 5, 6**), implicating the charged residues in the extended loop in the

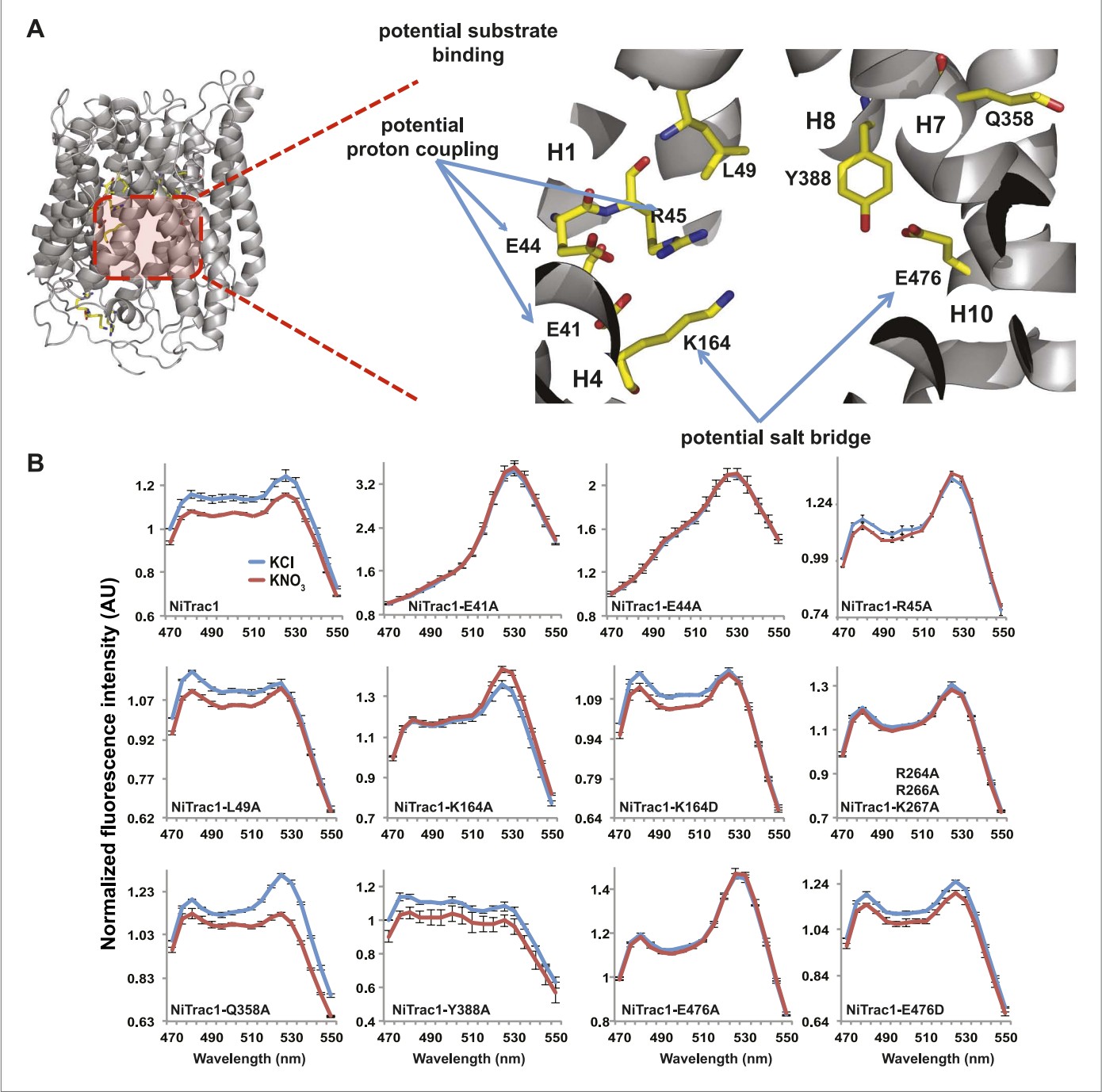

**Figure 4**. Response of NiTrac1 mutants to nitrate addition. (**A**) Three-dimensional model of CHL1 protein based on the crystal structures of bacteria ('Materials and methods'). Red square, potential substrate binding pocket. Right panel, enlarged potential substrate binding pocket. (**B**). Fluorescence response of NiTrac1 mutants expressed in yeast in response to addition of 10 mM potassium nitrate (red; control 10 mM KCl, blue; excitation at 428 nm). To compare the differences in fluorescence intensity between wild type and mutants of CHL1 as well as the differences after addition of nitrate, all data from wild type and mutants were normalized to the intensity of KCl-treated controls at 470 nm. Mean ± SD; n = 3.

regulation of nitrate uptake affinity. Similarly, the corresponding NiTrac1 mutant also lost the sensor response to high nitrate concentrations (**Figure 4B**). Together, these data show that NiTrac1 is a sensitive tool for reporting conformational changes in mutants and further support the hypothesis that NiTrac1 reports activity states of the transporter.

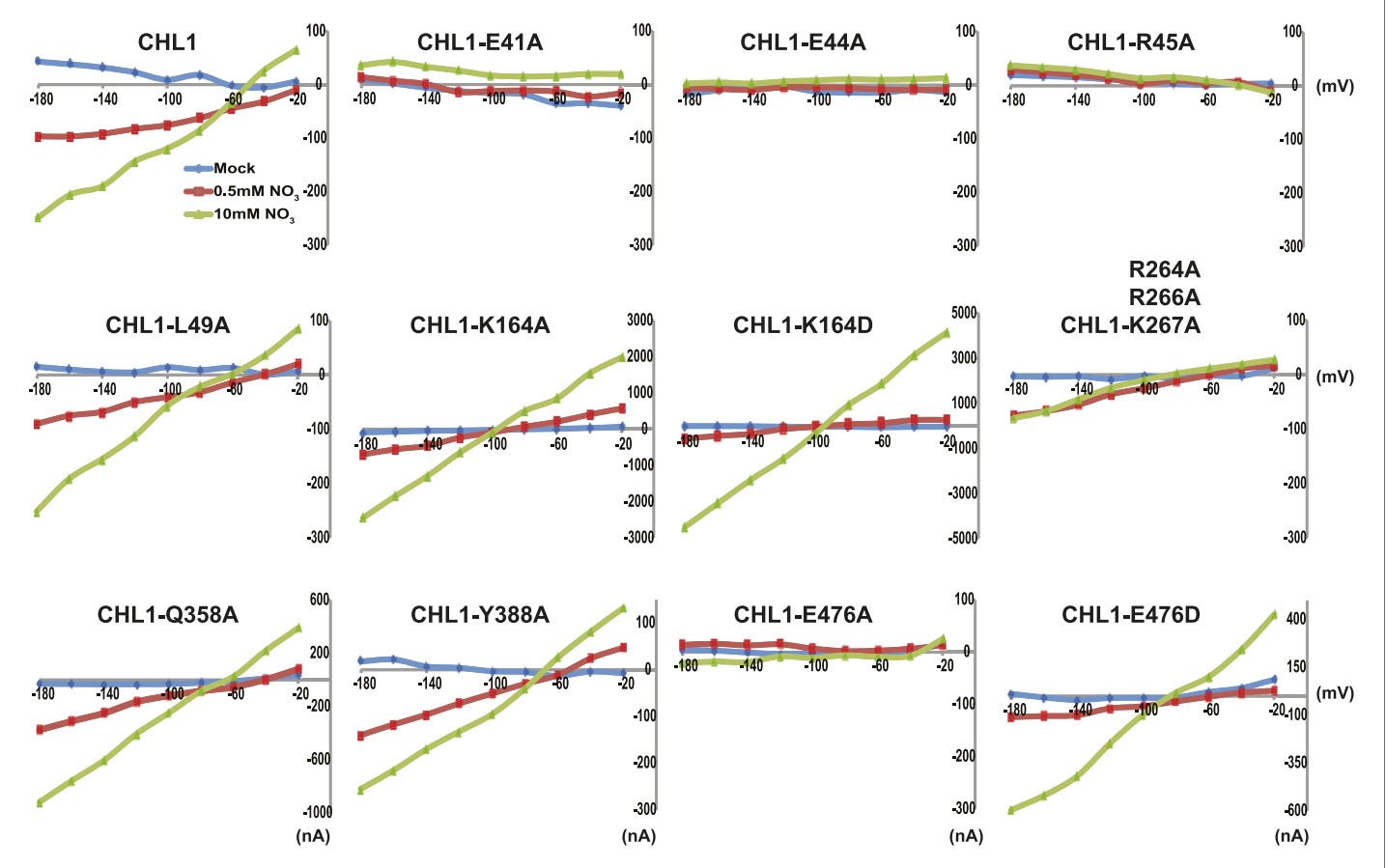

**Figure 5**. Current and voltage curve of CHL1/NRT1.1 mutants using TEVC. Oocytes were voltage clamped at −40 mV and stepped into a test voltage between −20 and −180 mV for 300 ms, in −20-mV increments. The currents (I) shown here are the difference between the currents flowing at +300 ms in the cRNA-injected CHL1 mutants and water-injected control of the indicated substrates. The curves presented here were recorded from a single oocyte. Similar results were obtained using another two different batches of oocytes.

## Effect of regulatory proteins on the NiTrac1 response

The transceptor CHL1 plays important roles in nitrate uptake, transport, sensing, and must therefore be subject to regulation of its activity by posttranslational regulation on the one hand; on the other hand, CHL1 must interact with intracellular proteins in order to control downstream transcription by signaling pathways. We hypothesized that binding of regulatory proteins or signaling proteins might affect the fluorescence properties of NiTrac1. Therefore, we tested whether coexpression of the known interactor CIPK23, which can phosphorylate CHL1 at T101 in in vitro assays, would affect the properties of NiTrac1 (*Figure 8*). CIPK23 did not change the energy transfer between the fluorophores in the absence of nitrate, but blocked the fluorescence response of NiTrac1 to nitrate addition (*Figure 8B*), either by stoichiometric binding or by phosphorylation of T101. The coactivator CBL9, which did not affect CHL1 transport activity on its own but enhanced the CIPK23-mediated phosphorylation of CHL1 (*Ho et al., 2009*), had no detectable effect on the fluorescence response of NiTrac1 by itself (*Figure 8C*). By contrast, CIPK8, which is nitrate inducible in a CHL1-dependent fashion, did not affect the Nitrac1 response. However, CBL1 on its own also blocked the Nitrac1 response to nitrate addition (*Figure 8D*). The analysis of coexpression of NiTrac1 with combinations of CIPKs and CBLs will require a different approach since episomal expression of three partners likely will create high variability due to copy number variance.

A large-scale interactome screen recently identified novel CHL1 interactors (*Lalonde, 2010*). To test whether some of these interactors affect NiTrac1 fluorescence, we coexpressed four candidate proteins with NiTrac1 in yeast. Although two of the four did not show significant effects on NiTrac1 or

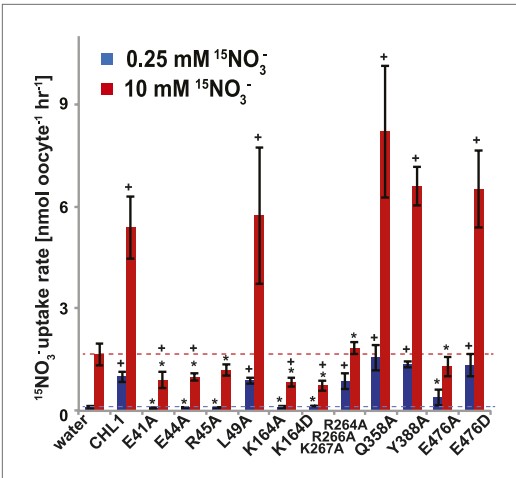

**Figure 6**. $^{15}NO_3^-$ uptake activity of various CHL1 mutants in oocytes. The injected oocytes with various cRNA of CHL1 mutants were incubated with 0.25 mM and 10 mM K$^{15}NO_3$ buffer at pH 5.5 for about 1.5~2 hr, and their $^{15}N$ content was determined as described in 'Materials and methods'. The values are mean ± SD (n = 5~6 for all three experiments). Data are normalized to the 0.25 mM treated CHL1-injected oocytes. +, significant difference (p<0.05, $t$ test) compared with water-injected oocytes. An asterisk indicates a significant difference (p<0.05, $t$ test) compared with the CHL1-injected oocytes. Similar results were obtained using another two batches of oocytes.

the response to nitrate addition, we found that the potassium transporter KT2 and the WNK kinase WNK8 blocked NiTrac1 responses (*Figure 8D*). Further experiments will be required to characterize the role of these new interactions; however, the results demonstrate the suitability of NiTrac1 for analyzing the effect of known and novel interactors on CHL1 conformation and activity.

## Discussion

To be able to monitor the activity and regulation of individual isoforms of the nitrate and peptide transporter family *in planta* and to study their structure function relationships, we engineered five transporters of the NPF/POT family to report their activity and conformation in vivo. The five proteins were fused with yellow and cyan versions of GFP at their N- and C-termini, respectively. When expressed in yeast, the sensors respond to substrate addition either by donor quenching or by a FRET change. The most striking feature of NiTrac is the biphasic response kinetic that matches the dual-affinity transport properties of the protein strikingly well. Based on a predicted structural model and sequence alignments, we mutated select amino acids. Analysis of the fluorescence response of these mutants and comparison with transport assays provides us with insights into the structure-function relationship of the CHL1 nitrate transceptor. The sensor is also used for probing structural rearrangements that occur when NiTrac is coexpressed with putative regulators and interactors, and we discovered new candidate regulatory proteins. The engineering of a suite of nitrate and peptide transport activity sensors complements our recent work in which we developed the prototype for fluorescence-based activity sensors AmTrac and MepTrac by inserting circularly permutated EGFP into conformation-sensitive positions of ammonium transporters. Addition of ammonium to yeast cells expressing the AmTrac/MepTrac sensors triggered concentration-dependent and reversible changes in fluorescence intensity (*De Michele et al., 2013*). Together, the engineering of activity sensors through two different approaches: insertion of a fluorophore into a conformationally sensitive site in AMT/MEPs and terminal fusions of a fluorophore pair to the NPF/POT family proteins indicate the potential to transfer the concept to other transporters, receptors, and enzymes. This suite of genetically encoded sensors provides a unique set of tools for observing the activity of individual transporter family members in intact tissue layers of intact plants.

### Sensor output from NiTrac1 and PepTracs

The activity sensors can provide three types of reports: (i) the basic ratio provides information on structure, specifically conformation of the population of sensors that can be compared between, for example, mutants or in response to coexpression of a regulator; (ii) the intensity of donor or acceptor can be subject to substrate-induced changes that lead to quenching as seen in NiTrac and PepTrac. At present, we do not understand the molecular basis of nitrate-induced donor quenching, which appears to affect mCerulean and CFP, but not mTFP. The fact that three PepTracs show a similar quenching effect when dipeptides are added may indicate that the quenching is caused by a conformational rearrangement in the transporter. A more detailed biophysical characterization may shed light on this unexpected behavior of the sensors. (iii) The change in the emission ratio of the two fluorophores upon substrate addition in PepTrac4 is likely caused by a change in the resonance energy transfer as had been observed for small molecule sensors (*Okumoto, 2012*). In certain cases, that is, NiTrac1 mutants L49A

| CHL1 | 15N | TEVC | FRET* | SR |
|---|---|---|---|---|
| WT | HA ✔ / LA ✔ | HA ✔ / LA ✔ | ↑ | DQ |
| E41A | HA ✗ / LA ✗ | HA ✗ / LA ✗ | ↑↑↑↑ | No DQ |
| E44A | HA ✗ / LA ✗ | HA ✗ / LA ✗ | ↑↑↑ | No DQ |
| E45A | HA ✗ / LA ✗ | HA ✗ / LA ✗ | ↑ | No DQ /iET |
| L49A | HA ✔ / LA ✔ | HA ✔ / LA ✔ | ↓ | DQ /iET |
| T101A | HA ✗ / LA ✔ | HA n.d. / LA n.d. | ↑ | DQ |
| K164A | HA ✗ / LA ✗ | HA ✔ / LA ✔ | ↑↑ | No DQ /iET |
| K164D | HA ✗ / LA ✗ | HA ✔ / LA ✔ | ↑ | DQ /iET |
| R264A/R266A/K267A | HA ✔ / LA ✗ | HA ✔ / LA ✗ | ↑↑ | No DQ |
| Q358A | HA ✔ / LA ✔ | HA ✔ / LA ✔ | ↑ | DQ /rET |
| Y388A | HA ✔ / LA ✔ | HA ✔ / LA ✔ | ↑ | DQ |
| E476A | HA ✗ / LA ✗ | HA ✗ / LA ✗ | ↑↑ | No DQ |
| E476D | HA ✔ / LA ✔ | HA ✔ / LA ✔ | ↑ | DQ |

**Figure 7**. Summary of nitrate uptake and TEVC measurements for CHL1 mutants and relative fluorescence emission and change in apparent energy transfer efficiency of NiTrac1 mutants in presence and absence of nitrate. CHL1 column: wild type and various mutants of CHL1. 15N column: nitrate uptake measured in oocytes using $^{15}NO_3^-$ (checkmark ✔ indicates transport activity detected, whereas ✗ indicates no significant, or dramatically reduced, uptake activity; red HA: measured at low nitrate concentration to analyze high-affinity component; blue LA: measured at high nitrate concentration to analyze low-affinity component). TEVC column: effect of mutations on current voltage relationships measured by two-electrode voltage clamping; checkmark ✔ indicates nitrate-induced current observed, whereas ✗ indicates no significant, or dramatically reduced, current induced by nitrate; red HA and blue LA as defined above. FRET* column: crude classification of the apparent energy transfer efficiency observed in NiTrac and NiTrac mutants in the absence of substrate; multiple arrows (↑) indicate a relative higher energy transfer efficiency. Blue arrows, reduced apparent energy transfer; red arrows, increased apparent energy transfer. SR column: type of response of NiTrac or mutants to substrate addition: DQ, donor quenching; iET, increased energy transfer; rET, reduced energy transfer.

and Q358A (*Figure 7*), we observed a mixture of donor quenching and FRET changes. We thus hypothesize that both NiTracs and all four PepTracs have the potential to report in two different modes, that is, donor quenching, a FRET change or a combination thereof.

Structural rearrangements triggered by mutations, by binding of a regulator or by mutations apparently lead to a variety of changes in the fluorescence output. One of the most striking features of NiTrac1 is that it reflects the biphasic kinetics of CHL1 and that even the transport and fluorescence response constants are highly similar. Mutagenesis of T101 to alanine, which had been shown to specifically affect the high-affinity component of nitrate uptake, also specifically eliminated the high-affinity response in NiTrac1. These findings strongly supported the notion that NiTrac reports the processes that occur in the transceptor, when it binds and/or transports nitrate. The observations also intimate that the dual-affinity is not caused by partial phosphorylation of CHL1 when expressed in oocytes, as suggested by Tsay's group (*Liu and Tsay, 2003*), but more likely represent an intrinsic property of CHL1 since they also occur when NiTrac1 is expressed in yeast. CHL1 also functions as nitrate sensor to regulate transcription of a variety of genes including that of the high-affinity nitrate transporter NRT2 (*Ho et al., 2009*). Interestingly, the transport and signaling activities of CHL1 can be decoupled by Pro492L in the loop connecting TM10 and 11 (*Ho et al., 2009*). It will thus be interesting to introduce this mutation into NiTrac1 and monitor the effect on the sensor output.

## Using NiTrac1 as a tool for structure function analyses

Taking the advantage of a homology model, we introduced mutations into NiTrac1 and studied the effects on the transport activity by TEVC recording and $^{15}$N-uptake into oocytes, and compared the effects to fluorescence readouts from the corresponding NiTrac1 mutants. Specifically, we analyzed the role of the putative proton-coupling motif 41ExxER45; the role of charged residues in the extended loop R264R266K267; and residues in the substrate binding pocket as well as a predicted a salt bridge L49, K164, Q358, Y388, and E476 (*Newstead, 2011*; *Newstead et al., 2011*; *Solcan et al., 2012*; *Doki et al., 2013*). We observed three main types of response (*Figure 7*): (i) loss of both nitrate uptake activity and loss of the sensor response in E41A, E44A, R45A, and E476A; (ii) loss of either high- or low-affinity uptake activity and correlated loss of the respective sensor response in T101A and R264A/R266A/K267A; and (iii) maintenance of the nitrate uptake activity and sensor response in L49A, Y388A, Q358A, and E476D. Relative to NiTrac1, more than half of the mutants show a change in FRET between the two fluorophores in the absence of substrate addition. Consistent with the role of E41 and E44 in

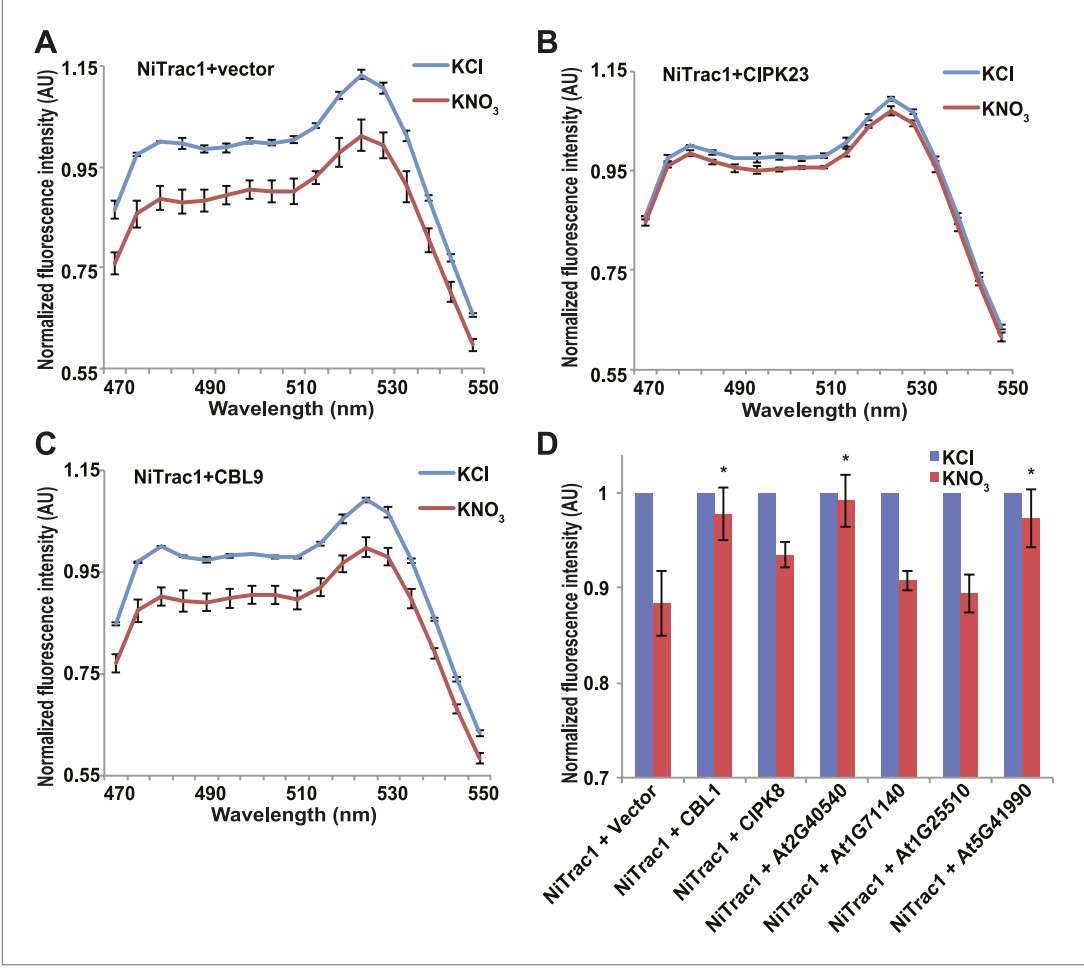

**Figure 8**. Effects of the fluorescence response of NiTrac1 by interacting proteins. Known interactors or regulators, such as CIPK8, CIPK23, CBL1, and CBL9 as well as other interactors identified in a large-scale membrane protein interaction screen were co-expressed with NiTrac1 in yeast cells. (**A**) Donor quenching response of NiTrac1 with vector as control. (**B**) and (**C**) Fluorescence response of NiTrac1 in CIPK23 and CBL9 coexpressing yeast, respectively. (**D**) The fluorescence response indicates that CIPK23, CBL1, At2g40540, and At5g41990 affect the conformation NiTrac1, whereas no detectable change is observed for CBL9, CIPK8, At1g71140, and At1g25510. Experiment performed as in *Figure 1C*. (**A–C**) Nitrate-induced ratio change (peak fluorescence intensity of Aphrodite excited at 505 nm over emission spectrum obtained with excitation at 428 nm). Data are normalized to KCl-treated control at 470 nm. An asterisk indicates a significant difference ($p<0.05$, *t* test) compared with the KNO$_3$-treated control. Mean ± SD; n = 3.

proton coupling for bacterial peptide transporters, the fluorescence response of E41A, E44A, and R45A NiTrac1 variants support a similar role in the nitrate transporter CHL1. Interestingly, L49A, which did not show detectable differences in transport activity, showed a mixture of donor quenching and FRET change in response to nitrate addition, demonstrating that NiTrac1 is exquisitely sensitive for detecting changes in the overall protein conformation. R45A lost transport activity but retained a FRET change response after the addition of substrate rather than showing a quenching response, indicating an overall conformational change due to binding of nitrate in the absence of a functional transport cycle. Interestingly, mutation of charged residues in the extended cytoplasmic loop of CHL1 (R264A/R266A/K267A) specifically affected the low-affinity component in sensor and uptake response, implicating the loop, potentially through interacting proteins that can tune activity. How T101 phosphorylation, which affects the behavior of CHL1/NiTrac1 at low nitrate levels cooperates with the cytosolic loop, which appears to specifically affect the behavior in high nitrate conditions will be interesting to

address in future experiments. Based on our studies, we presume that K164, Q358, and E476 may participate in nitrate binding. Consistent with data from bacterial peptide transporters, E476A lost both sensor response and uptake activity. This conserved residue aspartate likely plays a role in the binding pocket and/or salt bridge formation that is important for the substrate transport cycle. Mutants carrying K164A/D and Q385A mutations were both characterized by significantly increased nitrate-dependent inward currents. It will be interesting to further explore the cause for the increased conductivity with respect to transported ion species. Although the data from the limited number of mutants do not allow us to draw conclusions on the exact molecular nature of the conformational changes, we nevertheless provide the first evidence that activity sensors are highly sensitive and simple tools for probing structure-function relationships in heterologous and homologous systems without the necessity to purify the transporters.

## Effect of CIPK and CBL proteins on the NiTrac1 sensors

The interaction of proteins likely affects the conformation of both partners, either directly or as a consequence of modifications such as phosphorylation. Here we show that activity sensors can be used to probe such interactions with exquisite sensitivity. As a proof of concept, we demonstrate that coexpression of the calcium-dependent kinase CIPK23, which is phosphorylating T101 of CHL1 and thereby inhibiting the low-affinity component of CHL1, can block the fluorescence response when coexpressed with NiTrac1. Although typically CIPKs are thought to require a CBL for substrate recognition and derepression of the autoinhibition, CIPK23 had been shown to be able to interact with CHL1 on its own and trigger at least partial phosphorylation of T101 in vitro (*Ho et al., 2009*). Interestingly, although CBL9 had been shown to enhance the CIPK-mediated phosphorylation of T101, we did not observe an effect of coexpression of CBL on NiTrac1. Surprisingly, and despite the high sequence identity between AtCBL9 and AtCBL1 (~89% identity), CBL1 but not CBL9 inhibited the nitrate response of NiTrac1. AtCBL1 and 9 have been shown to regulate a variety of processes including potassium uptake, pollen germination, as well as sugar-, hormone- and ROS-signaling (*Sagi and Fluhr, 2006*; *Xu et al., 2006*; *Cheong et al., 2007*; *Hashimoto and Kudla, 2011*; *Drerup et al., 2013*; *Kimura et al., 2013*; *Li, 2013a*). Even though CBL1 and 9 have apparent overlapping functions, they can have specific effects, for example, the AtCBL1-AtCIPK1 complex is involved in ABA-dependent stress responses, whereas the AtCBL9-AtCIPK1 complex plays roles in ABA-independent stress responses (*Drerup et al., 2013*). In general, CIPKs depend on their coactivator-CBLs to activate CIPK kinase activity. However, recent studies showed that full-length CIPK23, CIPK16, or CIPK6 alone can activate the AKT1 potassium channel system (*Li et al., 2006*; *Lee et al., 2007*; *Fujii et al., 2009*). Also, AtCBL10 interacts with AKT1 to regulate potassium homeostasis without binding to any AtCIPKs (*Ren et al., 2013*). The assays deployed here use strong promoters and high copy number plasmids. It will therefore be important to test whether low levels of the kinase are sufficient for inhibiting NiTrac1. It will also be interesting to compare the responses of NiTrac1 when expressed in mutant plants lacking components of the CBL-CIPK machinery.

## WNK kinase and potassium transporter interactions

In addition, we tested whether NiTrac1 can be used to monitor conformational rearrangements caused by interacting proteins, specifically we tested interactors detected in a large-scale membrane protein/signaling protein interaction screen (*Lalonde et al., 2010*). Surprisingly, we found an interaction of CHL1 with the potassium transporter AtKT2/KUP2/SHY3, which plays a role in potassium uptake. Coexpression of KT2 with NiTrac1 led to a block of the nitrate response. Whether this interaction plays a role in crosstalk between nitrogen and potassium uptake remains to be shown. In addition, we had found an interaction with the '*no lysine (K) kinase 8*' WNK8. Also, WNK8 blocked the nitrate-induced fluorescence response of NiTrac1. WNK8 had been shown to interact specifically with and phosphorylate subunit C of the vacuolar $H^+$-ATPase AtVHA-C (*Hong-Hermesdorf et al., 2006*), as well as with the calcineurin B-like 1 calcium sensor AtCBL1 (*Li, 2013b*). It will be interesting to further explore the network between CBL1, WNK8, and CHL1.

Obviously, NiTrac1 is highly sensitive to conformational changes that occur during the transport cycle, effects of mutations, and to changes caused by interaction with other proteins. Thus, analyses performed with these sensors in plants will have to differentiate between responses caused by substrate and regulatory interactions. The use of controls, for example, the mTFP sensor and elimination

of FRET by exchanging the acceptor with a non-fretting fluorophore, as well as the use of mutant sensors may be a way to dissect the relative contribution of substrate and protein interactions. These new tools are complementary to the classical tools set including electrophysiology and tracer studies but have the clear advantage of allowing measurements deep inside plant or animal tissues and organs, domains largely inaccessible to other technologies.

## Outlook

In summary, we developed a set of five sensors that can report the activity of nitrate and peptide transporters in vivo. At the same time, such activity sensors prove to be sensitive tools for studying the effect of mutations on the conformation of the transporter or to detect the regulatory interactions with other proteins. The next step will be to deploy NiTrac1 and its mutants as well as the PepTracs in Arabidopsis plants to characterize the activity of the transporters and their regulation in vivo. The plant peptide transporters are close homologs of the human SLC15 peptide transporters. The SLC15 transporter PepT1 has pathophysiological relevance in processes like intestinal inflammation and inflammatory bowel disease (*Ingersoll et al., 2012*), and it serves as a key transport mechanism for uptake of drugs (*Agu et al., 2011*). Given the success in engineering five members of the plant transporter family, we envisage that the approach can be implemented also for measuring the activity of the human transporters in situ and to use such sensors, for example, for drug screens.

# Materials and methods

## DNA constructs

All transporter and sensor constructs were inserted by Gateway LR reactions, into the yeast expression vectors pDRFlip30, 34, and 39. pDRFlip30 is a vector that sandwiches the insert between an N-terminal Aphrodite t9 (AFPt9) variant (*Deuschle et al., 2006*), with nine amino acids truncated of C-terminus and a C-terminal monomeric Cerulean (mCer) (*Rizzo et al., 2006*). pDRFlip39 sandwiches the inserted polypeptide between an N-terminal enhanced dimer Aphrodite t9 (edAFPt9) and C-terminal fluorescent protein enhanced dimer, seven amino acids and nine amino acids truncated of N-terminus and C-terminus of eCyan (t7.ed.eCFPt9), respectively. pDRFlip34 carries an N-terminal AFPt9 and a C-terminal t7.TFP.t9 (t7.TFP.t9) (*Rizzo et al., 2006*). All plasmids contain the f1 replication origin, a GATEWAY cassette (attR1-*CmR-ccdB*-attR2), positioned between the pair of fluorescent proteins, the PMA1 promoter fragment, an ADH terminator, and the *URA*3 cassette for selection in yeast. Vector construction has been described (*Jones, In press*). The full length ORF of CHL1, PTR1, PTR2, PTR4, and PTR5 from Arabidopsis and different mutants of NRT1.1 in the TOPO GATEWAY Entry vector were used as sensory domains for creating the nitrate sensor NiTrac1 and the peptide sensors PepTrac1, PepTrac2, PepTrac4, and PepTrac5. The yeast expression vectors were then created by GATEWAY LR reactions between different forms of pTOPO-NRT/PRT and different pDRFlip-GWs, following manufacturer's instructions. For functional assays in *Xenopus* oocytes, the cDNAs of CHL1 and all mutants of CHL1 were cloned into the oocyte expression vector pOO2-GW (*Loqué et al., 2009*). Point mutations for studying characterization of CHL1 in oocyte and NiTrac1 in yeast were generated by QuikChange Lightning Site-Directed Mutagenesis Kit (Agilent Technologies, Santa Clara, CA). For the coexpression assays with interactors in yeast, putative interactors were inserted, by LR reaction, in the yeast expression vector pDR-XN-GW vector, which replaced *URA3* with *LEU2* in pDRf1 containing the f1 replication origin, GATEWAY cassette (-attR1-*CmR-ccdB*-attR2), PMA1 promoter fragment, and ADH terminator in yeast (*Loqué et al., 2007*).

## Yeast cultures

The yeast BJ5465 [MATa, *ura3–52*, *trp1*, *leu2Δ1*, *his3Δ200*, pep4::HIS3, prb1Δ1.6R, can1, GAL+] was obtained from the Yeast Genetic Stock Center (University of California, Berkeley, CA). Yeast was transformed using the lithium acetate method (*Gietz et al., 1992*), and transformants were selected on solid YNB (minimal yeast medium without nitrogen; Difco) supplemented with 2% glucose and *-ura/-ura-leu* DropOut medium (Clontech, Mountain View, CA). Single colonies were grown in 5 ml liquid YNB supplemented with 2% glucose, and *-ura/-ura-leu* drop out under agitation (230 rpm) at 30°C until $OD_{600nm}$ ~ 0.5 was reached. The liquid cultures were subcultured by dilution to $OD_{600nm}$ 0.01 in the same liquid medium and grown at 30°C until $OD_{600nm}$ ~ 0.2.

## Fluorimetry

Fluorimetric analyses are described in more detail at Bio-protocol (**Ho and Frommer, 2016**). In brief, fresh yeast cultures ($OD_{600nm}$ ~0.2) were washed twice in 50 mM MES buffer, pH 5.5, and resuspended to $OD_{600nm}$ ~0.5 in the same MES buffer supplemented with 0.05% agarose to delay cell sedimentation. Fluorescence was measured in a fluorescence plate reader (M1000; TECAN, Austria), in bottom reading mode using a 7.5 nm bandwidth for both excitation and emission (**Bermejo et al., 2010; Bermejo et al., 2011**). Typically, emission spectra were recorded ($\lambda_{em}$ 470–570 nm). To quantify fluorescence responses of the sensors to substrate addition, 100 µl of substrate (dissolved in MES buffer, pH 5.5 as 500% stock solution) was added to 100 µl of cells in 96-well flat bottom plates (#655101; Greiner, Monroe, NC). Fluorescence from cultures harboring pDRFlip30 (donor: mCER) and 39 (donor:t7.ed.eCFPt9) was measured by excitation at $\lambda_{exc}$ 428 nm; cell expressing from pDRFlip34 (donor t7.TFP.t9) was excited at $\lambda_{exc}$ = 440 nm.

Quantitative fluorescence intensity data from individual yeast cells expressing the sensors (**Figure 3C**) were acquired on an inverted microscope (Leica, Wetzlar, Germany). To be able to record fluorescence intensities in single cells over time, yeast cells were trapped as a single cell layer in a microfluidic perfusion system (Y04C plate, Onyx, Cellasic, Hayward, CA, USA) and perfused with either 50 mM MES buffer, pH 5.5, or buffer supplemented with 10 mM $KNO_3$ (**Bermejo et al., 2010**; **Bermejo et al., 2011**). Briefly, imaging was performed on an inverted fluorescence microscope (Leica DMIRE2) with a QuantEM digital camera (Photometrics) and a 40×/NA (numerical aperture) 1.25–0.75 oil-immersion lens (IMM HCX PL Apo CS). Dual-emission intensity ratios were simultaneously recorded using a DualView unit with a Dual CFP/YFP-ET filter set (ET470/24m and ET535/30m; Chroma) and Slidebook 4.0 software (Intelligent Imaging Innovations). Excitation (filter ET430/24x; Chroma) was provided by a Lambda LS light source (Sutter Instruments; 100%lamp output). Images were acquired within the linear detection range of the camera at intervals of 20 s. The exposure time was typically 1000 ms with an EM (electron-multiplying) gain of 3 °ø at 10 MHz and an electron multiplying charge coupled device (EMCCD) camera (Evolve, Photometrics, Tucson, AZ, USA). Measurements were taken every 10 s, with 100 ms exposure time using Slidebook 5.4 image acquisition software (Intelligent Imaging Innovations, Denver, CO, USA). Fluorescence pixel intensity was quantified using Fiji software; single cells were selected and analyzed with the help of the ROI manager tool.

## Structure prediction for CHL1 and PTR1

Protein structure prediction for CHL1 and PTR1 was performed using Phyre (**Kelley and Sternberg, 2009**). Full-length CHL1 (At1g12110) and AtPTR1/NPF8.1 (At3g54140) amino acid sequences were used for the 3D structure prediction on the website. The analysis made use of four solved crystal structures of nitrate/peptide homologs (PDB ID: 4iky, 2xut, 4aps, 4lep) (**Newstead, 2011**; **Newstead et al., 2011**; **Solcan et al., 2012**; **Doki et al., 2013**). The homologs shared 16–27% identity with CHL1 or PTR1. The predicted potentially functionally important residues were from the predicted structure (3DLigandSite, [**Wass et al., 2010**]) and from sequence alignments. After structural prediction of CHL1, 41-ExxER-45, in TM1, the conserved sequence motif involved in proton cotransport (22-ExxER-26, 21-ExxER-25, 21-ExxER-25, 32-ExxER-36 in $PepT_{St}$, $PepT_{So}$, $PepT_{So2}$, and GtPOT, respectively), putative residues involved in substrate binding pocket L49 in TM1 (Y30, Y29, Y29, and Y40 in $PepT_{St}$, $PepT_{So}$, $PepT_{So2}$, and GtPOT, respectively), Q358 in TM7 (Q289, Q317, Q291, and Q311 in $PepT_{St}$, $PepT_{So}$, $PepT_{So2}$, and GtPOT, respectively), and Y388 in TM8 (N328, N321, and N342 in $PepT_{St}$, $PepT_{So2}$, and GtPOT, respectively), and putative residues of salt bridges K164 in TM4 (K126, K127, K121, and K136 in $PepT_{St}$, $PepT_{So}$, $PepT_{So2}$, and GtPOT, respectively), E476 in TM10 (E400, E419, E402, and E413 in $PepT_{St}$, $PepT_{So}$, $PepT_{So2}$, and GtPOT, respectively), and residues R264/R266/K267 in the lateral helices loop between TM6 and TM7 were selected for mutagenesis.

## Functional expression of CHL1 and respective mutants in *Xenopus* oocytes

TEVC in oocyte was performed essentially as described previously (**De Michele et al., 2013**). In brief, for in vitro transcription, pOO2-CHL1 and respective mutants were linearized with *Mlu*I. Capped cRNA was in vitro transcribed by SP6 RNA polymerase using mMESSAGE mMACHINE kits (Ambion, Austin, TX). *Xenopus laevis* oocytes were obtained from the laboratory of Miriam Goodman by surgery manually or ordered from Ecocyte Bio Science (Austin, TX). The oocytes were injected via the Roboinjector (Multi Channel Systems, Reutlingen, Germany; [**Lemaire et al., 2004**; **Pehl et al., 2004**]) with distilled

water (50 nl as control) or cRNA from CHL1 or CHL1 mutants (50 ng in 50 nl). Cells were kept at 16°C 2–4 days in ND96 buffer containing 96 mM NaCl, 2 mM KCl, 1.8 mM CaCl$_2$, 1 mM MgCl$_2$, and 5 mM HEPES, pH 7.4, containing gentamycin (50 μg/μl) before recording experiments. Recordings were typically performed at day three after cRNA injection.

### Electrophysiological measurements in *Xenopus* oocytes

Electrophysiological analyses of injected oocytes were performed as described previously (*Huang et al., 1999*; *De Michele et al., 2013*). Reaction buffers used recording current (I)-voltage (V) relationships were (i) 230 mM mannitol, 0.3 mM CaCl$_2$, and 10 mM HEPES and (ii) 220 mM mannitol, 0.3 mM CaCl$_2$, and 10 mM HEPES at the pH indicated plus 0.5 or 10 mM CsNO$_3$. Typical resting potentials were ~−40 mV. Measurements were recorded by oocytes that were voltage clamped at −40 mV and a step protocol was used (−20 to −180 mV for 300 ms, in −20 mV increments) and measured by the two-electrode voltage-clamp (TEVC) Roboocyte system (Multi Channel Systems) (*Pehl et al., 2004*; *Lemaire et al., 2004*).

### $^{15}NO_3^-$ uptake assays in *Xenopus* oocytes

Nitrate uptake assays were performed using $^{15}$N-labeled nitrate (*Ho et al., 2009*), and oocytes injected with CHL1 cRNA were used as positive controls. After 2–4 days cRNA injection, the oocytes were incubated for 90~120 min in $^{15}NO_3^-$ medium containing 230 mM mannitol, 0.3 mM CaCl$_2$, 10 mM HEPES, and pH 5.5. Then, oocytes were rinsed five times with ND96 buffer and individually dried at 80°C for 1–2 days. $^{15}$N content was analyzed in an ECS 4010 Elemental Combustion System (Costech Analytical Technologies Inc., Valencia, CA, USA) whose output was connected to a Delta plus Advantage mass spectrometer (Thermo Fisher Scientific Inc., Waltham, MA, USA).

### Statistical analyses

For statistical analyses of $^{15}$N-nitrate uptake into oocytes (*Figure 6*) and the effect of treatments on the fluorescence responses (*Figures 1C and 8D*), we used analysis of deviance (ANOVA); factors (sample, treatment) were treated as fixed factors. ANOVAs were performed using the analysis of variance (ANOVA) calculator—one-way ANOVA from Summary Data (www.danielsoper.com/statcalc). All experiments were performed at least with three biological repeats. The reported values represent mean and standard deviation. Student's *t* test was used in *Figures 1, 6 and 8* to determine significance.

## Acknowledgements

We are very grateful to Yi-Fang Tsay for providing constructs and raw data for CHL1 nitrate uptake mediated by CHL1 into oocytes (*Figure 3*) and Alexander Jones for providing pDRFLIP vectors. Dr Ari Kornfeld is gratefully acknowledged for analyzing all $^{15}$N levels in oocytes shown here. We thank Drs Newstead and Parker (Oxford University) for providing access to their pre-publication data on the structure of CHL1 and for discussions regarding the transport mechanism. We thank Miriam Goodman (Stanford University) for providing *Xenopus* oocytes. We are grateful to Juan Simón Álamos Urzúa for technical assistance in PepTrac cloning and analysis.

## Additional information

### Funding

| Funder | Grant reference number | Author |
| --- | --- | --- |
| National Science Foundation | MCB-1021677 | Wolf B Frommer |
| National Science Foundation | MCB-1052348 | Wolf B Frommer |

The funder had no role in study design, data collection and interpretation, or the decision to submit the work for publication.

### Author contributions

C-HH, WBF, Conception and design, Acquisition of data, Analysis and interpretation of data, Drafting or revising the article

## Additional files

### Major datasets

The following previously published datasets were used:

| Author(s) | Year | Dataset title | Dataset ID and/or URL | Database, license, and accessibility information |
| --- | --- | --- | --- | --- |
| Doki S, Kato HE, Solcan N, Iwaki M, Koyama M, Hattori M, Iwase N, Tsukazaki T, Sugita Y, Kandori H, Newstead S, Ishitani R, Nureki O | 2013 | Crystal structure of peptide transporter POT (E310Q mutant) in complex with sulfate | 4IKY; http://www.rcsb.org/pdb/explore/explore.do?structureId=4iky | Publicly available at RCSB Protein Data Bank (http://www.rcsb.org). |
| Newstead S, Drew D, Cameron AD, Postis VL, Xia X, Fowler PW, Ingram JC, Carpenter EP, Sansom MSP, McPherson MJ, Baldwin SA, Iwata S | 2011 | Crystal structure of a proton dependent oligopeptide (POT) family transporter | 2XUT; http://www.rcsb.org/pdb/explore/explore.do?structureId=2xut | Publicly available at RCSB Protein Data Bank (http://www.rcsb.org). |
| Solcan N, Kwok J, Fowler PW, Cameron AD, Drew D, Iwata S, Newstead S | 2012 | Crystal structure of a POT family peptide transporter in an inward open conformation | 4APS; http://www.rcsb.org/pdb/explore/explore.do?structureId=4aps | Publicly available at RCSB Protein Data Bank (http://www.rcsb.org). |
| Guettou F, Quistgaard EM, Tresaugues L, Moberg P, Jegerschold C, Zhu L, Jong AJ, Nordlund P, Low C | 2013 | Structural insights into substrate recognition in proton dependent oligopeptide transporters | 4LEP; http://www.rcsb.org/pdb/explore/explore.do?structureId=4lep | Publicly available at RCSB Protein Data Bank (http://www.rcsb.org). |

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
