## [Decision Letter]

Thank you for sending your work entitled “Fluorescent sensors for activity and regulation of the nitrate transceptor CHL1/NRT1.1 and oligopeptide transporters” for consideration at *eLife*. Your article has been favorably evaluated by a Senior editor (Detlef Weigel), a Reviewing editor, and 2 reviewers, one of whom, Oliver Einsle, has agreed to reveal his identity.

The Reviewing editor and the reviewers discussed their comments before we reached this decision, and the Reviewing editor has assembled the following comments to help you prepare a revised submission.

In this study, the authors constructed the FRET-based nitrate sensor NiTrac1 by sandwiching CHL1 in between of two fluorescent proteins. NiTrac1 showed quenching of fluorescence in dependence of nitrate addition, and the quenching response showed biphasic kinetics like CHL1. The quenching was affected specifically by nitrate, by amino acid substitutions of CHL1 and interacting factors of CHL1. To control for a general quenching effect of nitrate on this type of reporter, the authors also constructed PepTracs, FRET-based peptide sensors constructed on the basis of PTR-type oligopeptide transporters. From the analysis of fluorophore-dependent emissions, the expression of cytosolic fluorophores and cyan fluorophore variants, the authors concluded that NiTrac1 and most PepTracs do not allow FRET-dependent but fluorophore-dependent substrate interaction to record in vivo activities of these membrane transporters.

A homology model based on a peptide transporter structure was used to design point-specific mutants. This part of the study is very well carried out and yields interesting results, to the point that the authors postulate the involvement of certain residues – K134, Q358 and E476 – in nitrate transport of NiTrac. This is a very nice result, as it implicated that the fusion approach can be used for functional studies of transport mechanism, but throughout the work the authors do not refer to two recent structures of bacterial nitrate transporters of the MFS family, NarK and NarU, that may be of high relevance here. While the reviewers were not aware of the degree of homology between these and CHL1, it might add substantially to the quality of the discussion to consider this new structural data within the present work (Yan et al. (2013) CellRep 3:716, and Zheng et al. (2013) Nature 481:469).

The experiments in this study appear to have been conducted with great care and the conclusions are consistent with the results. This study thus is complementary to a recent report from the same group, in which a permutated GFP was introduced into the cytosolic loop of ammonium transporters to record substrate-dependent conformational changes of the protein (14). Even though the authors are not yet able to explain the molecular mechanism underlying substrate-dependent fluorescence quenching, this study opens up a new avenue to monitor in vivo activities of transporters and thus promises to be highly influential to this field.

The following points should be addressed in a revised version of the manuscript:

1) One major concern that may require additional experimentation is the lack of a data set that shows time-dependent activation and reversibility of substrate-dependent fluorescence quenching. Such a data set is required not only to proof that activity and not activation is monitored by this molecular reporter but also to support its suitability to report in-planta changes of nitrate transport activities, i.e. when nitrate reduction depletes cytosolic nitrate pools. Such basic data are important as long as we don't know exactly what the quenching means. If these experiments cannot be done, the reasons for omitting them should be stated in the Discussion.

2) Figure 1: It would be highly useful to see changes in emission ratio after addition of potassium chlorate (not just chloride) since chlorate is a proven substrate analog for CHL1.

The reason to try chlorate is to obtain additional information on the responsiveness of the transporter-fluorophore complex to a substrate analog. If chlorate yields a similar response as nitrate while the other substrates did not, then I would consider this as an additional argument for a substrate-transporter interaction being responsible for the fluorescence quenching. (The reason to propose trying out chlorate is to obtain additional information on the responsiveness of the transporter-fluorophore complex to a substrate analog. If chlorate yields a similar response as nitrate while the other substrates did not, then one would consider this as an additional argument for a substrate-transporter interaction being responsible for the fluorescence quenching. Since one expects this to be a short-time response, there are no concerns about the toxicity of chlorate.)

---

## [Author Response]

*In this study, the authors constructed the FRET-based nitrate sensor NiTrac1 by sandwiching CHL1 in between of two fluorescent proteins. NiTrac1 showed quenching of fluorescence in dependence of nitrate addition, and the quenching response showed biphasic kinetics like CHL1. The quenching was affected specifically by nitrate, by amino acid substitutions of CHL1 and interacting factors of CHL1. To control for a general quenching effect of nitrate on this type of reporter, the authors also constructed PepTracs, FRET-based peptide sensors constructed on the basis of PTR-type oligopeptide transporters. From the analysis of fluorophore-dependent emissions, the expression of cytosolic fluorophores and cyan fluorophore variants, the authors concluded that NiTrac1 and most PepTracs do not allow FRET-dependent but fluorophore-dependent substrate interaction to record in vivo activities of these membrane transporters*.

*A homology model based on a peptide transporter structure was used to design point-specific mutants. This part of the study is very well carried out and yields interesting results, to the point that the authors postulate the involvement of certain residues – K134, Q358 and E476 – in nitrate transport of NiTrac. This is a very nice result, as it implicated that the fusion approach can be used for functional studies of transport mechanism, but throughout the work the authors do not refer to two recent structures of bacterial nitrate transporters of the MFS family, NarK and NarU, that may be of high relevance here. While the reviewers were not aware of the degree of homology between these and CHL1, it might add substantially to the quality of the discussion to consider this new structural data within the present work (Yan et al. (2013) CellRep 3:716, and Zheng et al. (2013) Nature 481:469)*.

*The experiments in this study appear to have been conducted with great care and the conclusions are consistent with the results. This study thus is complementary to a recent report from the same group, in which a permutated GFP was introduced into the cytosolic loop of ammonium transporters to record substrate-dependent conformational changes of the protein (*[14]*). Even though the authors are not yet able to explain the molecular mechanism underlying substrate-dependent fluorescence quenching, this study opens up a new avenue to monitor in vivo activities of transporters and thus promises to be highly influential to this field*.

My understanding is that is had been contentious whether CHL1 (NRT1) and PTRs are MFS transporters until the POT structures came out. The bacterial NARs, for which structures have been generated seem to be more closely related to the second class of plant nitrate transporters, called NRT2. The Yan et al. (2013) structure paper on NarU states that it is related to the NRT2 family. The homology between NarU and NRT2s is extremely low: 14%–18%. We did a forced ‘alignment’ of CHL1 with NarU and NARK and find identities of 11 and 13%, which would appear random. Therefore, we did not see a good way to address this comment.

*The following points should be addressed in a revised version of the manuscript*:

*1) One major concern that may require additional experimentation is the lack of a data set that shows time-dependent activation and reversibility of substrate-dependent fluorescence quenching. Such a data set is required not only to proof that activity and not activation is monitored by this molecular reporter but also to support its suitability to report in-planta changes of nitrate transport activities, i.e. when nitrate reduction depletes cytosolic nitrate pools. Such basic data are important as long as we don't know exactly what the quenching means. If these experiments cannot be done, the reasons for omitting them should be stated in the Discussion*.

We have performed two sets of experiments to address this point. We trapped yeast cells in a microfluidic device and monitored time-dependent changes. These data are now shown as Figure 3. As you can see, we observed coincident and rapid mCerulean quenching upon addition of nitrate (Aphrodite, when excited directly is unaffected like in the fluorimeter assays). The response is concentration-dependent. Importantly, the response is reversible, again coincident with removal and rapid. The experiment has been performed three times independently with comparable results. These data now strongly support that the sensors will likely work well when implemented in plants. We agree that this is a significant improvement.

*2)*Figure 1*: It would be highly useful to see changes in emission ratio after addition of potassium chlorate (not just chloride) since chlorate is a proven substrate analog for CHL1*.

*The reason to try chlorate is to obtain additional information on the responsiveness of the transporter-fluorophore complex to a substrate analog. If chlorate yields a similar response as nitrate while the other substrates did not, then I would consider this as an additional argument for a substrate-transporter interaction being responsible for the fluorescence quenching. (The reason to propose trying out chlorate is to obtain additional information on the responsiveness of the transporter-fluorophore complex to a substrate analog. If chlorate yields a similar response as nitrate while the other substrates did not, then one would consider this as an additional argument for a substrate-transporter interaction being responsible for the fluorescence quenching. Since one expects this to be a short-time response, there are no concerns about the toxicity of chlorate*.*)*

We analyzed the effect of chlorate as requested. We would however like to note that we have not been able to find a single manuscript that has tested the effect of chlorate on CHL1, nor its paralogs in plants nor orthologs from other plant species. That is indeed odd, given that CHL stands for chlorate resistance. It is noteworthy that Arnold Bloom (Kosola and Bloom, 1996), one of the prominent physiologists in this field, and others have reported that inhibition of nitrate uptake by nitrate was not competitive, therefore questioning the mechanism of chlorate transport. These experiments were performed in planta, where the situation is likely complex. Bloom concluded that chlorate is not a good analog for tracer studies in planta. Nevertheless, we tested the effect of chlorate on NiTrac1 and observe a similar quenching response, even quantitatively, for chlorate as observed for nitrate. These data are now shown in the new panel as Figure 1 (subsequent figure sections are the same, just renumbered). Thus the new data address the point raised by the reviewers.